# Long-COVID in Patients with Cancer Previously Treated with Early Anti-SARS-CoV-2 Therapies in an Out-of-Hospital Setting: A Single-Center Experience

**DOI:** 10.3390/cancers15041269

**Published:** 2023-02-16

**Authors:** Angioletta Lasagna, Giuseppe Albi, Simone Figini, Sara Basile, Paolo Sacchi, Raffaele Bruno, Paolo Pedrazzoli

**Affiliations:** 1Medical Oncology Unit, Fondazione IRCCS Policlinico San Matteo, 27100 Pavia, Italy; 2Department of Electrical, Computer and Biomedical Engineering, University of Pavia, 27100 Pavia, Italy; 3Division of Infectious Diseases I, Fondazione IRCCS Policlinico San Matteo, 27100 Pavia, Italy; 4Department of Clinical Surgical Diagnostic and Pediatric Sciences, University of Pavia, 27100 Pavia, Italy; 5Department of Internal Medicine and Medical Therapy, University of Pavia, 27100 Pavia, Italy

**Keywords:** real-life data, long-COVID, vaccine, cancer, SARS-CoV-2, outcome, brain fog, remdesivir, nirmatrelvir/ritonavir

## Abstract

**Simple Summary:**

This article regards the incidence of long COVID symptoms in a cohort of patients with cancer with or without previous treatment with early therapies anti-SARS-CoV-2 in an out-of-hospital setting. The enrolled patients were invited to take part in the survey by telephone at least 12 weeks after COVID-19 diagnosis in order to evaluate the incidence of long COVID symptoms. To date, no papers have focused on the oncological population managed at home with the early anti-SARS-CoV-2 therapies. The overlap between the symptoms related to the oncological disease/oncological treatment and the symptoms of long COVID is one of the main future challenges that oncologists will have to manage.

**Abstract:**

The incidence of long COVID in a cohort of patients with cancer with or without previous treatment with early therapies anti-SARS-CoV-2 in an out-of-hospital setting have to be elucidated. We prospectively enrolled all patients treated for a solid tumor at the department of Medical Oncology of the Fondazione IRCCS Policlinico San Matteo with a positive SARS-CoV-2 antigen or polymerase chain reaction test from January to September 2022 (Omicron surge). Ninety-seven patients answered the survey questions by telephone at least 12 weeks after COVID-19 diagnosis in order to evaluate the incidence of long COVID symptoms. Only twelve patients (12.4%) reported long COVID. No significant difference between early therapies anti-SARS-CoV-2 31 and long COVID (*p* = 0.443) was seen. The female sex (*p* = 0.024) and diabetes mellitus (*p* = 0.014) are significantly associated with long COVID. No statistically significant difference between the two groups (Long COVID vs. No Long COVID) according to the time to nasal swab viral clearance (*p* = 0.078). The overlap between the symptoms related to the oncological disease/oncological treatment and the symptoms of long COVID is one of the main future challenges that oncologists will have to manage.

## 1. Introduction

Almost three years after the emergence of the severe acute respiratory syndrome coronavirus 2 (SARS-CoV-2) [1], the Coronavirus Disease 2019 (COVID-19) pandemic still represents a challenge for all the healthcare professionals. Since the beginning of the pandemic, the immunocompromised patients, such as those with cancer, have been considered to be at higher risk of infection with poor prognosis. One of the first comprehensive meta-analyses of studies about the outcome of COVID-19 patients with cancer demonstrated that the fatality rate was higher in the patients with cancer compared to that of patients without cancer [2]. The development of vaccines has changed the natural history of the pandemic with the consequent reduction of the rate of severe COVID-19 disease. In particular, the third vaccine dose of COVID-19 vaccine induces a strong immune response against the ancestral D614G variant and against SARS-CoV-2 Variants of Concern (VOCs), such as Omicron, also in the patients with cancer [3,4].

Despite the effectiveness of COVID-19 vaccines having been in real world, many reports suggest the occurrence of breakthrough infections (BTIs) even in a vaccinated population. In particular, a systematic review reported that BTIs are more common in immunocompromised individuals than in fully vaccinated healthy individuals [5]. New options to treat high-risk outpatients with mild-to-moderate COVID-19 are represented by the use of first-generation oral antiviral agents against SARS-CoV-2 (such as nirmatrelvir/ritonavir and molnupinavir) and parenteral therapeutics, including anti-spike monoclonal antibodies and remdesivir [6]. Their clinical effects were seen in unvaccinated patients, and the trials took place before the emergence of the Omicron variant. The real-world effectiveness of monoclonal antibodies and oral antiviral agents in patients with cancer with COVID-19 is still largely uncharacterized. In our previous paper, we measured the time to negative SARS-CoV-2 respiratory sample and the symptoms’ duration in a cohort of COVID-19 patients treated with the available early therapies and in a cohort of untreated COVID-19 patients. We demonstrated that a higher percentage of treated patients had a reduced time to negative sample (73% vs. 18%, *p* = 0.0011) and shorter symptoms’ duration (94% vs. 27%; *p* < 0.0001) compared to the untreated patients [7].

Whether the acute phase of COVID-19 appears to be currently controlled by vaccines and early therapies, there is growing evidence of prolonged long-term effects, called long COVID syndrome as far as 12 months after the onset of the initial COVID-19 diagnosis. According to the definition of the World Health Organization (WHO), long COVID is “A condition which occurs in individuals with a history of probable or confirmed SARS-CoV-2 infection, usually three months from the onset of COVID-19 with symptoms that last for at least two months and cannot be explained by an alternative diagnosis” [8].

Long COVID is characterized by a multidimensional symptomatology and disability, and the predominant symptoms are fatigue, headache, brain fog and myalgia [9,10]. The diagnostic challenge is to distinguish symptoms attributed to long COVID from those of a pre-existing or otherwise unrelated disease to COVID-19. Long COVID is common and may persist for at least 2 years after SARS-CoV-2 infection; even severe cases are rarely reported [9]. However, some symptoms (such as brain fog and fatigue) can be disabling, and significantly impact daily activities and interpersonal relationships [10]. Up to 37% of patients can present persistent clinical symptoms in the 90 to 180 days after diagnosis [11].

Moreover, the impact of long COVID symptoms in cancer patients is not so easy to assess, because they can typically report during systemic anticancer treatment the most common symptoms such as fatigue or brain fog. To date, few reports have investigated the prevalence, duration and clinical significance of long COVID specifically in the patients with cancer [12,13], and all these articles reported data collected before the Omicron surge, before the COVID-19 vaccine and without considering the available outpatient COVID-19 therapies.

The aim of this study is to evaluate these aspects in the patients with cancer who experienced long COVID symptoms and to compare them with those who did not.

## 2. Materials and Methods

### 2.1. Participants

The patients analyzed in this paper originated in an ambispective observational 102 cohort study on all the cancer patients treated at the department of Medical Oncology of the Fondazione IRCCS Policlinico San Matteo with a positive SARS-CoV-2 antigen or polymerase chain reaction test from January to September 2022. We considered only the outpatients in order to avoid bias related to severe disease. The infectious disease specialist chose the most appropriate drug among sotrovimab, molnupiravir, remdesivir and nirmatrelvir/ritonavir.

The study was conducted in accordance with the Declaration of Helsinki and approved by the local Ethics Committee (Comitato Etico Area Pavia) and Institutional Review Board (protocol code P-0039959/22). All the subjects signed an informed written consent.

### 2.2. Outcomes

Our primary outcome was to evaluate the rate of hospitalization for COVID-19 disease within 14 days in the patients with cancer using early anti-SARS-CoV-2 therapies as per indication. Preliminary results about the real-world effectiveness of monoclonal antibodies and oral antiviral agents in preventing progression to severe COVID-19 in patients with cancer have been published elsewhere [7]. Our secondary outcomes were the incidence of long COVID symptoms according to WHO case definition, the time to COVID-19 symptoms resolution and the time to nasal swab viral clearance in the two groups (Long COVID vs. No Long COVID). In this article, the analyses focused on these purposes.

### 2.3. Data Collection

The enrolled patients were invited to take part in the survey by telephone at least 12 weeks after COVID-19 diagnosis in order to evaluate the incidence of long COVID symptoms. The survey consisted of a predefined list of the most common symptoms of long COVID symptoms: respiratory symptoms (breathlessness/cough), fatigue, anosmia, ageusia, hair loss, chest pain, palpitations, diarrhea, skin rashes, brain fog, headache, cough and myalgia. Oncologists conducted the telephone survey in order to discriminate as far as possible between the symptoms related to the oncological disease/oncological treatment and the long COVID symptoms.

We reported these patients’ characteristics: sex, age, comorbidities, type of solid cancer, type of oncological treatment, treatment setting (first- or second-line, maintenance after chemo-radiotherapy, neo/adjuvant setting), date of the onset of the symptoms, type and duration of symptoms, re-infections, type of early therapy and status of COVID-19 vaccination.

### 2.4. Statistical Analysis

The patients’ characteristics are described as median and interquartile range (Shapiro–Wilks test excluded the normal distribution hypothesis) if quantitative variables. Qualitative variables are described as count and percentages. Considering an expected long COVID proportion of 0.15, a confidence level of 95% and a margin of error of 7.4%, we calculate the sample size should be at least 90 patients. The width of the 95% exact binomial confidence interval for the long COVID prevalence will range from 7.6% to 22.4%. To compare continuous and categorical variables between long COVID and no long COVID groups, the Mann–Whitney-U test and Chi-square test or Fisher’s exact test were used, respectively. The time-to-event data were compared using the log-rank test. A multivariable logistic regression analysis was performed to investigate the associations between type of solid cancer and cancer stage, type of oncological treatment and treatment setting with the long COVID status. To exclude the effect of potential confounders, variables that reported a significant difference in the univariate analyses were included in the multivariable analysis. Backward variable selection based on the Akaike information criterion (AIC) was used to select the variables to include in the multivariable analysis. The multivariable analysis results are expressed in odds ratio (OR) and 95% CI, while discrimination performance is measured by the area under the receiver operating characteristic (AUC). All the statistical tests were performed two-sided, and a *p* value < 0.05 was considered statistically significant. Rstudio (R version 4.0.5.) was used for all the computations and analysis.

## 3. Results

### 3.1. Patients’ Characteristics

From January to September 2022, one hundred and three patients with cancer (62 females, 41 males; median age 63, inter-quartile range [IQR] 15.5) reported a laboratory-confirmed diagnosis of SARS-CoV-2 infection. Thirty-one patients had breast cancer (30.1%); twenty-eight (27.2%) had lung cancer; twenty-four (23.3%) had gastrointestinal cancer; eight (7.8%) had melanoma; four (3.9%) had kidney cancer; the remaining eight patients (7.7%) had other types of tumors. Fifty-three patients (51.5%) were on chemotherapy; twenty-eight (27.2%) were on immune checkpoint inhibitors (ICIs); and twenty-two (21.3%) received a targeted therapy.

Ninety-two patients (89.3%) were symptomatic for COVID-19 at the time of diagnosis (55 females, 37 males). Seventy-six (73.8%) received an early therapy anti-SARS-CoV-2 in an out-of-hospital setting. In particular, fifty-one patients (67.1%) received nirmatrelvir/ritonavir; sixteen patients (21.1%) received molnupinavir, while six patients (7.9%) and three patients (3.9%) received remdesivir and sotrovimab, respectively.

### 3.2. Statistical Analyses Comparing Long COVID vs. No Long COVID

At the time of the survey by telephone, six patients (6/103, 5.8%) had died for the oncological disease; the remaining ninety-seven patients accepted to answer the survey questions (Table 1).

Twelve patients (12.4%, median age 58, IQR 16) reported long COVID symptoms. The majority of them were females (11/12, 91.7%) (Figure 1).

In details, four reported myalgia (33.3%); seven (58.3%) and six (50%) patients indicated fatigue and brain fog, respectively. Three patients (25%) reported respiratory symptoms. Two patients (16.7%) had lung cancer; eight (66.7%) had breast cancer; one had melanoma; and the other patient had ovarian cancer. The majority of the patients were on chemotherapy (6/12, 50%); two patients (16.7%) were on ICIs; and four (33.3%) were on targeted therapy. All the patients were vaccinated with three doses of the mRNA vaccines, and they presented a symptomatic infection at the time of COVID-19 diagnosis. Eight patients (8/12, 66.7%) received an early anti-SARS-CoV-2 therapy: seven patients received nirmatrelvir/ritonavir, and one received remdesivir. Three patients did not receive the treatment because the symptoms appeared more than 5 days before, and one patient refused to receive the treatment.

Eighty-five patients (87.3%, median age 63, IQR 16) did not report long COVID symptoms (47/85, 55.3% females and 38/85, 44.7% males). Twenty-three patients (27.1%) had lung cancer; twenty-one (24.7%) had breast cancer; twenty-three (27.1%) had gastrointestinal cancer; seven (8.2%) had melanoma; four (4.7%) had kidney cancer; and seven (8.2%) had other types of cancer. The majority of the patients were on chemotherapy (43/85, 50.6%); twenty-five patients (29.4%) were on ICIs; and seventeen (20%) were on targeted therapy.

In the univariate analyses, we found no significant association between early treatment and long COVID (*p* = 0.443). Among the other considered variables in our cohort, only the female sex (*p* = 0.024), obesity (*p* = 0.039) and diabetes mellitus (*p* = 0.014) have been shown to relate to the occurrence of long COVID symptoms, while the other variables did not (Table 2).

In the multivariable analyses, we include the variables that were significant in the univariate analyses and the variables of interest for which we want to investigate the association with the long COVID status. Diabetes Mellitus, sex and breast cancer were selected by the backward variable selection, leading to an AIC = 61.4. The logistic regression model reached an AUC of 0.88. We showed that the diabetes mellitus (*p* = 0.004) is the only associated variable with the long COVID status (Table 3).

We found no statistically significant difference between the two groups (Long COVID vs. No Long COVID) according to the time to COVID 19 symptoms resolution (*p* = 0.064) and the time to nasal swab viral clearance (*p* = 0.078). (Figure 2).

## 4. Discussion

Our telephone survey of 97 cancer patients with a laboratory-confirmed diagnosis of SARS-CoV-2 infection found that only 12 patients (12.6%) reported long COVID symptoms. This percentage is in line with what other authors have reported [12].

The patients with long COVID had an increased antigen-specific CD4+ T cell response to the SARS-CoV-2 S protein, with a prolonged T cell response magnitude and an increased expression of PD-1-expressing T lymphocytes that indicated an exhaustion of T cells [14]. Queiroz and colleagues demonstrated higher levels of the pro-inflammatory cytokines (IL-17 and IL-2) in the patients with long COVID compared to the patients without [15]. This evidence seems to suggest the link between long COVID and the residual inflammation in the damaged organs [15]. So, patients with cancer who have an impairment of immune responses due to the oncological disease present an additive risk of long COVID symptoms [14]. Cancer itself and the oncological therapies are associated with increased levels of pro inflammatory cytokines [16]. Several cytokines such as interleukin-1 (IL-1) and interleukin-6 (IL-6) seem to play a key role in oncogenesis [17]. Moreover, some pro-inflammatory cytokines have implicated in a wide variety of cancer-related symptoms. For example, TNF-alpha and IL-1beta are involved in etiopathogenesis of the fatigue [18]. At the same time, there is growing evidence that pro-inflammatory cytokines are increased in long COVID patients, as reported above [15]. In our cohort, the stage disease was not associated statistically (*p* = 0.274) with long COVID. A confounding element may be the patient’s difficulty in distinguishing the symptoms of cancer from those of long COVID and thus ascribing them to the oncological disease rather than to the long COVID.

The role of the early therapies in the contest of long COVID is intriguing. Little data are currently available about this topic. In a prospective study [19], the authors observed that the antiviral treatment with remdesivir in the hospitalized COVID patients had a protective effect on the onset of long COVID symptoms (*p* < 0.001). These data might be probably due to the shorter time of viral replication, with a consequent reduction of the chronic inflammation. In addition, in a non-randomized, controlled trial in Shanghai, nirmatrelvir/ritonavir proved to reduce significantly the viral shedding. The subjects who received nirmatrelvir/ritonavir had a shorter viral shedding time [11.11 (2.67) vs. 9.32 (2.78), *p* = 0.001] compared to the control group [20]. In our cohort, no statistical difference between the two groups (long COVID vs. no long COVID) was highlighted based on the use of early anti-SARS-CoV-2 therapies. This might be due to the small number of patients with long COVID, rather than the lack of role of the early therapies. Another possibility might be that all the patients included in our study had mild COVID-19, and therefore the benefit of the early therapies in this setting might be less evident than in the setting of severe COVID-19.

In our cohort, all the patients with long COVID had received three doses of the mRNA vaccine. In the literature, the role of vaccination on the risk of long COVID is still unclear. In an observational cohort study in the United Kingdom, the long COVID symptoms were reported by 9.5% of the subjects double-vaccinated before COVID-19 infection and by 14.6% of the unvaccinated controls (adjusted odds ratio [OR], 0.59 [95% confidence interval (CI), 0.50–0.69]) [21]. In another cohort study conducted during the Delta (B.1.617.2) variant surge, no differences in long COVID symptoms were demonstrated between vaccinated and non-vaccinated groups six months after the diagnosis, but the vaccinated group reported worse sleep quality (*p* = 0.03) than the non-vaccinated group [22]. In our cohort, no statistical difference between the two groups (long COVID vs. no long COVID) was highlighted on the vaccination status, but the limited sample size obliges us not to generalize the results.

In our study, 11 patients (92%) with long COVID were female, and this variable is as a result statistically significant in the univariate analyses (*p* = 0.024). This evidence is in line with the available data in the literature. In a multicenter cohort study, the authors demonstrated that the female sex was significantly associated with ≥3 post-COVID symptoms (adjusted OR 2.54, 95% CI 1.671–3.865, *p* < 0.001) [23]. Similarly, Townsend and colleagues reported a higher prevalence of fatigue following COVID-19 infection in females than in males [24]. Several different mechanisms can explain why females are more prone to post-COVID symptoms than males. The male sex is associated with a higher risk of adverse outcomes during the acute phase of COVID-19 [25] and presented a longer time for virus shedding compared to women [26]. This could be related to the biological differences in the expression of angiotensin-converting enzyme-2 (ACE2) that is encoded by the ACE2 gene located on the X chromosome, and females may have a potentially more efficient form of ACE2 receptor [27]. Moreover, estrogens are able to suppress the levels of IL-6 by the alteration of CD16 expression and the influence of the levels of Natural Killer (NK) cells [28], and this may explain the higher percentage of long COVID among the females [23]. In addition, the microbiome could play a key role, similarly to that observed in the acute phase [29]. In a recent prospective study, Liu and colleagues investigated the gut microbiome composition of 106 patients from COVID-19 diagnosis up to six months later. The patients with long COVID presented a distinct gut microbiome dysbiosis with *Faecalibacterium prausnitzii*, which is able to inhibit the secretion of IL-6 and seems to have the largest inverse correlations with long COVID at 6 months [30].

Moreover, we have considered the comorbidities of our patients. The diabetes mellitus and obesity resulted statistically significant variables related to long COVID symptoms according to the data available in the literature [31,32]. A recent systematic literature review and meta-analysis evaluated the risk factors potentially predictive of the development of long COVID [33]. The authors highlighted that obesity was associated with longer persistence of symptoms, more risk of pathological pulmonary limitations and metabolic abnormalities, while they did not find any association between the diabetes mellitus and long COVID.

The different types of tumors and oncological therapies do not seem to be associated with an increased risk of long COVID, but future studies with larger case series will be useful to confirm or disprove these results. Immunological studies among subjects with long COVID who had mild acute COVID-19 (no hospitalization and no respiratory disease) have demonstrated T cells exhaustion with reduced CD4+ and CD8+ effector memory cell numbers and elevated PD1 expression on central memory cells [34]. The evaluation of these data in a larger sample of patients with cancer undergoing immunotherapy might be interesting and provide new information about the pathogenesis of long COVID.

Finally, we assessed whether patients with long COVID had a longer duration of symptoms and/or a longer duration of viral shedding during the acute phase of the disease. Several papers have suggested the viral persistence as a trigger for the symptoms of long COVID [34,35]. In our paper, we found no statistically significant difference between the two groups (long COVID vs. no long COVID) according to the duration of the COVID-19 symptoms of the acute phase and the time to nasal swab viral clearance. However, we only investigated the possible persistence of the virus through the nasopharyngeal swab and not in other ways (e.g., in urine, stool, plasma). Future research will be addressed to these issues.

The strength of our data consists in the prospective design and a well-defined cancer population treated for mild COVID-19 with early therapies anti-SARS-CoV-2 in an out-of-hospital setting during the Omicron surge. Moreover, we have collected all the information at the same time-point (12 weeks after the diagnosis of COVID-19).

The small sample size of the patients with long COVID is the main limitation of the study. Moreover, most symptoms are self-patient-reported symptoms prone to observer bias. The lack of validated scales to measure most of the symptoms and the extremely heterogeneity of them make difficult to compare data between subjects or studies. The number of the patients without early therapies anti-SARS-CoV-2 is lower than that of the other group, which may have reduced the statistical power. Finally, we have not conducted a randomized clinical trial (RCT), which is the only one that can guarantee a balanced distribution of measured and unmeasured confounders between those treated with early therapies anti-SARS-CoV-2 and those not treated with early therapies anti-SARS-CoV-2. We recognize that an RCT is difficult in clinical practice for ethical issues. Therefore, real-world studies appear even more relevant to assess the incidence of the symptoms of long COVID in cancer patients through time and to evaluate how these symptoms may influence cancer treatments and prognosis.

## 5. Conclusions

Female sex and the diabetes mellitus seem to be significantly related to long COVID symptoms in patients with cancer on active treatment, while no significant association was found between the early anti-SARS-CoV-2 therapies and long COVID. Future real-world data in larger cohort are warranted to confirm these results. The overlap between the symptoms related to the oncological disease/oncological treatment and the symptoms of long COVID is one of the main future challenges that oncologists will have to manage. Very few studies have specifically focused on long COVID symptoms in cancer patients undergoing active oncological treatment. It is important to make oncologists aware to follow the patients over time and to recognize long COVID symptoms promptly. We believe it may be useful to perform an evaluation at least 12 weeks after COVID-19 diagnosis in all cancer patients in order to intercept those symptoms potentially related to long COVID-19. Moreover, this paper suggests the need of training oncologists in recognizing long COVID symptoms and the need to create a network of multidisciplinary collaborations between the various healthcare professionals for the best management of cancer patients.

## Figures and Tables

**Figure 1 cancers-15-01269-f001:**
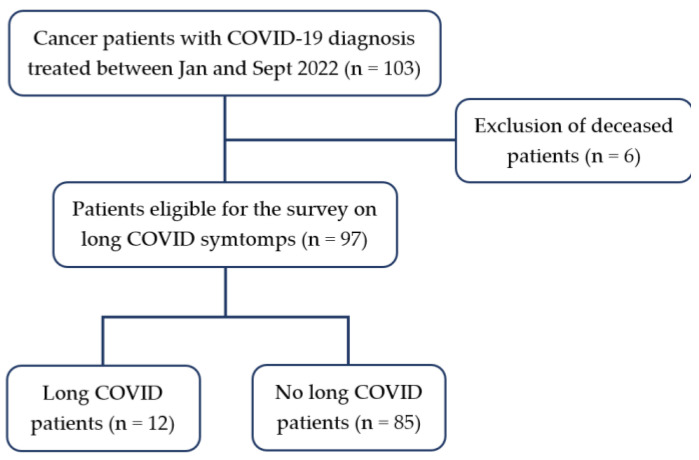
Flowchart.

**Figure 2 cancers-15-01269-f002:**
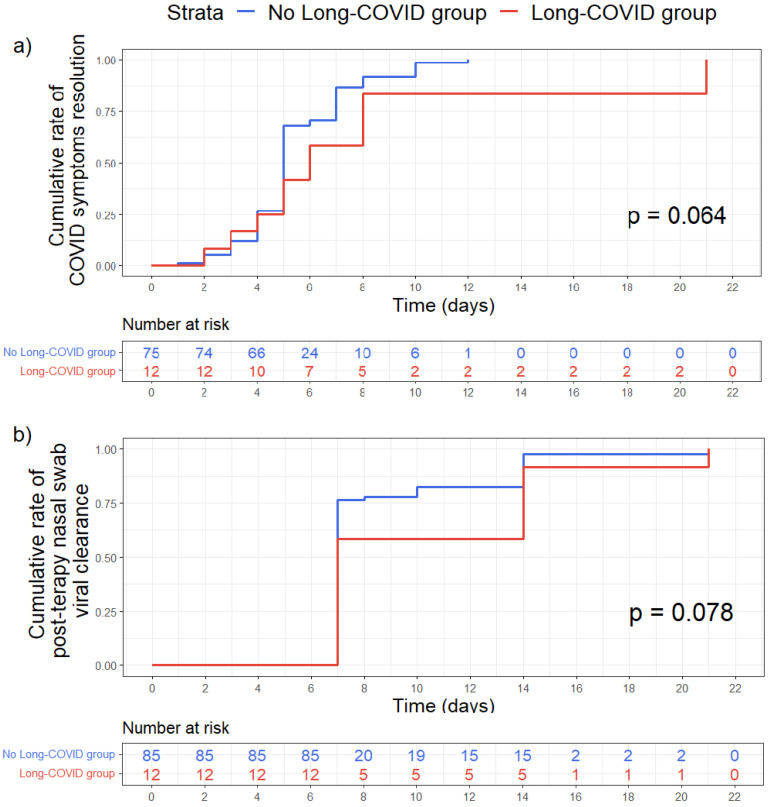
Kaplan–Meier curves comparing time-to-event between Long-COVID group (red) and No Long-COVID group (blue). (**a**) Time in days of COVID symptoms resolutions. (**b**) Time in days to obtain the first negative nasal swab after therapy. The reported *p* values resulted from Log-rank test. Time-to-event were assessed with Kaplan–Meier plot and compared with log rank test.

**Table 1 cancers-15-01269-t001:** Survey questions.

Symptoms	Yes	No
Respiratory symptoms (breathlessness/cough)		
Myalgia		
Palpitations		
Chest pain		
Fatigue		
Brain fog		
Headache		
Gastrointestinal symptoms (diarrhea/anorexia)		
Skin rashes		
Anosmia		
Ageusia		
Hair loss		

**Table 2 cancers-15-01269-t002:** Clinical and demographic characteristics of cancer patients enrolled with telephone follow-up (whole sample and grouped by presence of long COVID).

Variable	Whole Sample (n = 97)	Long COVID Group (n = 12, 12.4%)	No Long COVID Group (n = 85, 87.6%)	*p* Value
Age [years], median (IQR)	63 (16)	58 (16)	63 (16)	0.290
Sex, n (%)				**0.024**
Females	58 (60%)	11 (91.7%)	47 (55.3%)	
Males	39 (40%)	1 (8.3%)	38 (44.7%)	
Type of tumor, n (%)				0.076
Lung	25 (25.8%)	2 (16.7%)	23 (27.1%)	
Melanoma	8 (8.2%)	1 (8.3%)	7 (8.2%)	
Breast	29 (29.9%)	8 (66.7%)	21 (24.7%)	
Kidney	4 (4.2%)	0	4 (4.7%)	
Gastrointestinal	23 (23.7%)	0	23 (27.1%)	
Other	8 (8.2%)	1 (8.3%)	7 (8.2%)	
Stage of tumor, n (%)				0.274
II/III	24 (24.8%)	5 (41.7%)	19 (22.3%)	
IV	73 (75.2%)	7 (58.3%)	66 (77.7%)	
Type of oncological treatment, n (%)				0.475
ICIs	27 (27.8%)	2 (16.7%)	25 (29.4%)	
Chemotherapy **	49 (50.5%)	6 (50%)	43 (50.6%)	
Target therapy/ormonotherapy	21 (21.7%)	4 (33.3%)	17 (20%)	
CKD, n (%)	1 (1%)	0	1 (1.2%)	1
COPD, n (%)	17 (17.5%)	1 (8.3%)	16 (18.8%)	0.686
Diabetes mellitus type 2, n (%)	19 (19.6%)	6 (50%)	13 (15.3%)	**0.014**
Hypertension, n (%)	22 (22.7%)	3 (35%)	19 (22.4%)	1
Ischemic heart disease, n (%)	2 (2.1%)	0	2 (2.4%)	1
Obesity, n (%)	7 (7.2%)	3 (25%)	4 (4.7%)	**0.039**
Vaccination doses, n (%)				0.847
0	3 (3.1%)	0	3 (3.5%)	
1	1 (1%)	0	1 (1.2%)	
2	3 (3.1%)	0	3 (3.5%)	
3	81 (83.5%)	12 (100%)	69 (81.2%)	
4	9 (9.3%)	0	9 (10.6%)	
Time between vaccination dose and positivity [months], median (IQR)	7 (4)	7 (3)	7 (4)	0.991
Type of early therapies, n (%)				0.443
No treatment	25 (25.8%)	4 (33.3%)	21 (24.7%)	
Sotrovimab	2 (2%)	0	2 (2.3%)	
Molnupinavir	15 (15.5%)	0	15 (17.7%)	
Remdesivir	6 (6.2%)	1 (8.3%)	5 (5.9%)	
Nirmatrelvir/ritonavir	49 (50.5%)	7 (58.4%)	42 (49.4%)	
Symptoms’ duration [days], median (IQR)	5 (3)	6 (3)	5 (3)	0.212
First negative post-therapy swab [days], median (IQR)	7 (1)	7 (7)	7 (0)	0.124

Legend: IQR: interquartile range, ICIs: immune-checkpoints inhibitors, CKD: chronic kidney disease, COPD: chronic obstructive pulmonary disease. *p* values are reported in bold if significant. ** Type of chemotherapy: cisplatin and gemcitabine, FOLFOX, FOLFIRI, gemcitabine and Nab-paclitaxel, eribulin, docetaxel, epirubicin and cyclophosphamide, carboplatin and pemetrexed, gemcitabine, trastuzumab emtansine (T-DM1), pemetrexed.

**Table 3 cancers-15-01269-t003:** Multivariable analysis: Logistic regression for assessing association between variables and long COVID.

Outcome: Long COVID			
	Adjusted OR	95% CI	*p* Value
Diabetes Mellitus	10.03	2.24–55.32	**0.004**
Sex = Female	6.27	0.74–137.5	0.132
Breast cancer	4.43	0.98–26.76	0.069

Legend: OR: odd ratio; CI: confidence interval. *p* values are reported in bold if significant.

## Data Availability

All the data supporting the findings of this study can be found within the article.

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
