# Peer review of "Long-COVID in Patients with Cancer Previously Treated with Early Anti-SARS-CoV-2 Therapies in an Out-of-Hospital Setting: A Single-Center Experience"

_cancers, 2023, doi:10.3390/cancers15041269_

Round 1
Reviewer 1 Report
The association of symptoms between long COVID and cancer is one of the major challenges in field of cancer therapy. In this regard the author’s approach to elucidate this is a good initiative. The study evaluated the related aspects in cancer patients experiencing long COVID symptoms and compared them with those who did not.
Major Comments
1. The study highlights association between females, diabetes mellitus and long COVID symptoms in patients with cancer on active treatment. Hence it will be a good idea to highlight Type I or Type 2 diabetes mellitus is more prone for this. If possible, the relative age of the patients more susceptible for this observation.
2. Please include a discussion on the involvement of signaling pathway of cancer or hallmarks of cancer involved in this study, correlated to long COVID symptoms.
3. Please also include the probable strategy or approach by the oncologist for addressing the challenge of overlap between the symptoms related to cancer treatment and the symptoms of long COVID
Minor Comments
1. In Table 1 and Table 2 please include the name of the drug administered side by to “Chemotherapy”, if possible.
2. Please include a flowchart or graphical abstract describing the summary of this study.
Author Response
Major Comments
-
The study highlights association between females, diabetes mellitus and long COVID symptoms in patients with cancer on active treatment. Hence it will be a good idea to highlight Type I or Type 2 diabetes mellitus is more prone for this. If possible, the relative age of the patients more susceptible for this observation.
Author response: All of our patients had Type 2 diabetes mellitus. I added this information in the text.
2. Please include a discussion on the involvement of signaling pathway of cancer or hallmarks of cancer involved in this study, correlated to long COVID symptoms.
Author response: Long COVID symptoms and symptoms cancer-related show some similarities. We added in the discussion a short paragraph about the role of cytokines in cancer and long COVID patients.
3. Please also include the probable strategy or approach by the oncologist for addressing the challenge of overlap between the symptoms related to cancer treatment and the symptoms of long COVID
Author response: Very few studies specifically target long-COVID symptoms in cancer patients undergoing active oncological treatment. It is important to awaken the oncologist to follow the patient over time and to recognize long-COVID symptoms promptly.
We believe it may be useful to perform an evaluation at least 12 weeks after COVID-19 diagnosis in all cancer patients in order to intercept those symptoms potentially related to Long COVID-19.
Moreover, this paper suggests the need of training oncologists in recognizing long COVID symptoms and the need to create a network of multidisciplinary collaborations between the various healthcare professionals for the best management of cancer patients.
Minor Comments
-
In Table 1 and Table 2 please include the name of the drug administered side by to “Chemotherapy”, if possible.
Author response: I modified the table 1 adding these data.
2. Please include a flowchart or graphical abstract describing the summary of this study.
Author response: I added the figure 2 with the flowchart
Reviewer 2 Report
Abstract:
Please highliht: What is the underlying research question?
It is not correct to state that 97 patients among 103 agreed to take part in the survey. The 6 patients who had died were - of course - not able to answer these question. Considering this fact, the paticipation rate was 100%.
Introduction, lines 62 and 63: What do the percentages (73% versus 18%) and (94% versus 27%) indicate?
Outcomes: The primary outcome is the rate of hospitalization. However, the main aspect of this paper is the comparison between two subgroups "Long Covid" and "No long Covid".
Statistical analysis:
First sentence (line 159): the 6 patients didn't die at the time of the survey, they already HAD DIED.
Analyzing several variables simultaneously, is a "multivariable" or a "multiple analysis". The term "multivariate" is not correct.
Not the p values are 2-sided, but the statistical tests.
Table 2: In general, there is nothing wrong with perfoming a multiple analysis including a rather large numer of variables. However, authors shoud use a selection method (forward, backward or stepwise) in order to get a final model including only variables with a significant impact on the outcome. Furthermore, the AUC (Area under the curve) of the final model should be given.
Discussion: At the end of this chapter, authors suggest to conduct a RCT in order to compare treated and untreated patients. Theoretically, this is an interesting approach. However, it might be problematic for ethical reasons to conduct such an RCT. Authors should discuss this point. More than that: This is not the underlying research question of their manuscript.
Conclusions: Authors should highlight the new findings that have emerged from their study.
Author Response
Abstract:
Please highlight: what is the underlying research question?
It is not correct to state that 97 patients among 103 agreed to take part in the survey. The 6 patients who had died were - of course - not able to answer these question. Considering this fact, the participation rate was 100%.
Author response: I have modified the text of the abstract
Introduction:
lines 62 and 63: What do the percentages (73% versus 18%) and (94% versus 27%) indicate?
Author response: I modified the text.
Outcomes
The primary outcome is the rate of hospitalization. However, the main aspect of this paper is the comparison between two subgroups "Long Covid" and "No long Covid".
Author response: As reported in the paragraph 2.2, the primary outcome was the assessment of the rate of hospitalisation for COVID-19 disease. We have already reported in our previous paper the results about the efficacy of monoclonal antibodies and oral antiviral agents in preventing progression to severe COVID-19 in cancer patients. In this paper, we focused on secondary outcomes whose results were not available at the time of the first publication.
Statistical analysis
First sentence (line 159): the 6 patients didn't die at the time of the survey, they already HAD DIED.
Analyzing several variables simultaneously, is a "multivariable" or a "multiple analysis". The term "multivariate" is not correct.
Not the p values are 2-sided, but the statistical tests.
Author response: We corrected the text
Table 2: In general, there is nothing wrong with perfoming a multiple analysis including a rather large numer of variables. However, authors shoud use a selection method (forward, backward or stepwise) in order to get a final model including only variables with a significant impact on the outcome. Furthermore, the AUC (Area under the curve) of the final model should be given.
Author response: We corrected the table 2
Discussion
At the end of this chapter, authors suggest to conduct a RCT in order to compare treated and untreated patients. Theoretically, this is an interesting approach. However, it might be problematic for ethical reasons to conduct such an RCT. Authors should discuss this point. More than that: This is not the underlying research question of their manuscript.
Author response: An RCT is difficult in practice for ethical issues, so real-world studies appear even more important to assess the incidence of the symptoms of long COVID in cancer patients through time and how these symptoms may influence cancer treatments.
I added this concept in the discussion.
Conclusions
Authors should highlight the new findings that have emerged from their study.
Author response: In our study, we observed that only 12 patients (12.6%) reported long COVID symptoms. This percentage, although not high, should not be underestimated. The oncologist plays an important role in the correct management of patients and in the correct differential diagnosis between COVID-related symptoms and tumour-related symptoms. Very few studies specifically target long-COVID symptoms in cancer patients undergoing active oncological treatment. It is important to awaken the oncologist to follow the patient over time and to recognise long-COVID symptoms promptly. We believe it may be useful to perform an evaluation at least 12 weeks after COVID-19 diagnosis in all cancer patients in order to intercept those symptoms potentially related to Long COVID-19. Moreover, this paper highlights the need of training oncologists in recognizing long COVID symptoms and the need to create a network of multidisciplinary collaborations between the various healthcare workers for the best management of cancer patients.
Round 2
Reviewer 2 Report
All suggestions for improvement have been incorporated. In the present version I would like to recommend the publication.
Author Response
I have done the english language corrections according to the editors' suggestions.
